# Magnetic-Activated Cell Sorting (MACS): A Useful Sperm-Selection Technique in Cases of High Levels of Sperm DNA Fragmentation

**DOI:** 10.3390/jcm9123976

**Published:** 2020-12-08

**Authors:** Alberto Pacheco, Arancha Blanco, Fernando Bronet, María Cruz, Jaime García-Fernández, Juan Antonio García-Velasco

**Affiliations:** 1IVI Madrid, 28023 Madrid, Spain; Fernando.bronet@ivirma.com (F.B.); maria.cruz@ivirma.com (M.C.); Juan.Garcia.Velasco@ivirma.com (J.A.G.-V.); 2Science Faculty, Alfonso X “El Sabio” University, Villanueva de la Cañada, 28691 Madrid, Spain; 3IVI Salamanca, 37001 Salamanca, Spain; arancha.blanco@ivirma.com; 4The University of Edinburgh Medical School, Edinburgh EH8 9AG, UK; jaimegfvelasco@gmail.com; 5Rey Juan Carlos University, 28922 Madrid, Spain

**Keywords:** sperm DNA fragmentation, SDF, sperm selection, magnetic-activated cell sorting, MACS, male infertility, miscarriage, ART outcome

## Abstract

Magnetic-activated cell sorting (MACS) can be used to separate apoptotic sperm with high proportions of fragmented DNA from the rest, thus improving the overall quality of the seminal sample. Therefore, the aim of this retrospective study was to investigate the efficiency of the MACS technique to increase reproductive outcomes in patients with high levels of sperm DNA fragmentation (SDF) undergoing intracytoplasmic sperm-injection (ICSI) cycles. In this study, we analyzed a total of 724 assisted-reproduction-technique (ART) cycles that were divided into two groups: the study group (*n* = 366) in which the MACS selection technique was performed after density-gradient centrifugation (DGC), and the control group (*n* = 358) in which only DGC was used for sperm selection. Reproductive outcomes were analyzed in both groups according to three different ART procedures: preimplantation genetic testing for aneuploidy (PGT-A), and autologous and oocyte-donation cycles. The MACS group showed significantly lower miscarriage rates in autologous ICSI cycles, higher pregnancy rates in oocyte-donation cycles, and a significant increase in live-birth rates in both autologous and oocyte-donation cycles. Overall, these results suggested that the MACS technique can be effectively used to eliminate sperm with high SDF levels, and therefore may help to improve reproductive outcomes in couples undergoing ART.

## 1. Introduction

The success of assisted-reproduction techniques (ART) depends largely on the quality of the gametes used in fertility treatment cycles. However, in most cases, seminal quality is still analyzed using only conventional seminal analysis. Due to its variability and low specificity, this is not particularly effective in diagnosing functional alterations or in characterizing the origin of sperm defects [1].

One of the most promising tools to diagnose infertility in males is analysis of sperm DNA fragmentation [2]. Despite published results showing great variability, several meta-analyses confirmed that DNA integrity is crucial for appropriate embryo development [3,4], implantation, and ongoing pregnancy [5,6,7]. Alterations in the apoptotic processes, defects in DNA remodeling during spermatogenesis, and epididymal oxidative damage are the main mechanisms causing single- or double-strand DNA breaks in spermatocytes [8,9].

Apoptosis is a physiological process aiming to remove abnormal spermatozoa in order to maintain testicular homeostasis in terms of germ-cell population and testicular-nutrient availability [10]. Apoptotic mechanisms induce the activation of nuclear endonucleases that lead to the generation of double-strand breaks (DSBs) on sperm DNA. These are closely related with implantation failures and miscarriage [11]. Sertoli cells are able to distinguish and eliminate those sperm cells with apoptotic markers on the outer plasma membrane, such as phosphatidylserine (PS) residues. Functional alterations of Sertoli cells or an overactivity of testicular apoptotic processes (known as abortive apoptosis) can impair the elimination of apoptotic cells during spermatogenesis and consequently increase the number of apoptotic sperm cells in the ejaculate [12].

Under normal physiological conditions, apoptotic sperm cells with externalized PS residues on the plasma membrane can be recognized and efficiently eliminated in the female genital tract by phagocytes [13], preventing the fertilization of the oocyte by a spermatozoon with alterations in its DNA integrity [14]. This important sperm-selection process is inconveniently bypassed when ART sperm-selection techniques are performed, such as density-gradient centrifugation or sperm swim-up. This is because these techniques are only based on the recovery of sperm populations with high motile capacity. Moreover, the subsequent selection of the viable spermatozoon to be injected into an oocyte in intracytoplasmic sperm-injection (ICSI) cycles depends on the subjective preference of the embryologist based on the best sperm morphology. Thus, these processes do not include the physiological identification of nonfunctional and viable sperm in the female genital tract.

In order to address this issue, different sperm-selection techniques have recently been developed on the basis of different functional aspects and activation processes that happen during sperm-cell capacitation. One is the magnetic-activated cell-sorting (MACS) technique, which is used to identify and positively eliminate apoptotic cells from ejaculate [15] on the basis of the identification of externalized PS residues on apoptotic sperm cells by annexin V-conjugated superparamagnetic microbeads [16,17,18]. This technique also reduces the proportion of sperm with fragmented DNA in the ejaculate before using it for ART procedures [19]. In addition, there are no clinically adverse effects at the obstetric and perinatal levels in newborns who were closely monitored following oocyte-donation and sperm-selection treatments with MACS [20].

There are still conflicting results regarding the benefits of the MACS technique in ART. Some publications confirmed the MACS selection technique as a useful method to reduce the number of apoptotic sperm, thus improving overall embryo quality and pregnancy rates [21,22]. However, other authors did not observe significant differences between MACS and conventional sperm-selection techniques in reproductive outcomes [23,24]. Such discrepancies may be due to the great variability and low number of patients included in these studies. Most studies have not considered male-related factors in patient selection criteria or patient demographic analysis, such as levels of sperm DNA fragmentation. For this reason, in this retrospective study, we analyzed whether the MACS sperm-selection technique could improve the reproductive outcome in selected patients with increased rates of sperm DNA fragmentation.

## 2. Experiment Section

### 2.1. Study Population and Design

A total of 724 fertility-treatment cycles undergoing assisted reproductive treatment in our institution from 2015 to 2018 were included in this nonintervention retrospective cohort study of patients assessed with routine clinical examinations and procedures. All procedures were approved by our Institutional Review Board (1401-MAD-001-DA) and complied with the Spanish Law of Assisted Reproductive Technologies (14/2006).

Patients with severe male factor were referred for andrology evaluation in our institution prior to assisted reproductive treatment for a detailed investigation of male fertility and to rule out other treatable urological pathologies (i.e., orchitis, epididymitis, or varicocele). As inclusion criteria, only patients with high levels (>20% according to our internal threshold level and other previously published thresholds [25]) of sperm DNA fragmentation measured by a terminal deoxynucleotidyl transferase (TdT)-dUDP nick-end labeling (TUNEL) assay were included in the study (Figure 1). Exclusion criteria included semen samples with seminal infection (including *Mycoplasma hominis* and *Ureaplasma urealiicum* determination), orchitis/epididymitis, AZF microdeletions, altered karyotype, sperm aneuploidies determined by fluorescent in situ hybridization (FISH) analysis, and patients with systemic diseases or history of cryptorchidism. The choice of whether to perform MACS was based on a clinician’s judgment after patient counseling. Patients using their own oocytes, and those undergoing oocyte-donation treatment were included. Details of donor selection and work-up studies were previously reported [20].

### 2.2. Sperm DNA Fragmentation (SDF) Assessment

SDF was assessed with a TUNEL assay using an in situ Cell Death Detection Kit (Roche Diagnostics, Barcelona, Spain) and following the manufacturer’s instructions. Semen samples were washed with phosphate buffered saline (PBS; Gibco, Invitrogen, Barcelona, Spain) and centrifuged at 3000 rpm for 3 min. After discarding the supernatant, cells were fixed with a freshly prepared fixation solution (PBS + paraformaldehyde 1%) for 60 min at room temperature. Cells were then washed with PBS and incubated with permeabilization solution (0.1% Triton X-100 in 0.1% sodium citrate) for 2 min at 4 °C. After incubation, samples were washed twice with PBS and then incubated for 1 h at 37 °C with a TUNEL reaction mixture containing a TdT enzyme solution. Each experimental setup also included a positive control in which the sample was incubated with 3 IU/mL DNase I recombinant in 50 mmol/L Tris–HCl, pH 7.5 (Roche Diagnostics) for 15 min at 25 °C to induce DNA fragmentation prior to labeling procedures; and a negative control in which the sample was incubated with reaction mixture in the absence of the enzyme solution. Samples were washed twice in PBS for 3 s at 7200 g, and cells were lastly resuspended in a volume of 1 mL of PBS buffer. Sperm DNA fragmentation (SDF) was quantified using a flow cytometer (MACSQuant, Miltenyi Biotec, Teterow, Germany), and at least 20,000 cells were analyzed. Threshold value was set to 20% of sperm with fragmented DNA according to our previous data [25].

### 2.3. Male Evaluation and Conventional Sperm Selection

Semen samples were collected by masturbation into nontoxic sterile plastic jars after 3 to 5 days of sexual abstinence. Time of sperm collection was adjusted to the moment when oocytes had to be microinjected to minimize the possible effect of reactive oxygen species (ROS). Samples were liquefied for 30 min at room temperature (22 °C) and evaluated according to WHO criteria [26], including volume, pH, viscosity, and visual appearance. Then, the conventional density-gradient centrifugation (DGC) technique was performed for sperm selection. For DGC, ALLGrad (LifeGlobal, Guelph, ON, Canada) was diluted in medium for Global Fertilization (LifeGlobal, Guelph, ON, Canada) to obtain dilutions of 45 and 90%. Two gradient columns were prepared in Falcon tubes by gently layering 1 mL of each solution, starting with the 90% fraction at the bottom. One milliliter of the semen sample was stratified on top of the discontinuous gradient columns and centrifuged for 18 min at 300× *g*. After centrifugation, the obtained pellet was collected and washed twice at 350 g for 5 min. This sample was then used for an ICSI procedure (control group) or a MACS selection technique (study group) before the ICSI procedure.

### 2.4. MACS Sperm-Selection Technique

In the study group, after the swim-up procedure, a nonapoptotic selection procedure was performed in the samples by using the MACS ART Annexin V Reagent (Miltenyi Biotec, Teterow, Germany) and following manufacturer’s instructions. Briefly, obtained cells in the postgradient centrifugation were centrifuged at 300× *g* for 4 min. After the elimination of the supernatant, cells were resuspended in 100 mL of binding buffer. Then, the cellular suspension was incubated for 15 min at room temperature with the annexin-V reagent. The sperm/microbead suspension was loaded into a separation column previously rinsed with 1.5 mL of binding buffer and attached to a MACS magnet (MiniMACS, Miltenyi Biotec, Teterow, Germany). The positive fraction (apoptotic sperm) was retained into the separation column, while the negative fraction (containing nonapoptotic sperm) was eluted through the column and collected into a tube. Lastly, the negative fraction was centrifuged and, after discarding the supernatant, resuspended in fresh medium (Fert, Origio, Málov, Denmark) prior to the ICSI procedure.

### 2.5. Cycle Procedure and Outcome

Ovarian stimulation in patients and donors was carried out as previously described [27].

Serum β-hCG analysis was performed 12 days after embryo transfer (ET), and clinical pregnancy was confirmed when a gestational sac with a fetal heartbeat was visible in an ultrasound scan.

Fertilization rate (FR) was defined as the number of fertilized oocytes over the microinjected ones. Clinical outcomes included clinical pregnancy rate (PR), miscarriage rate (MR), and live-birth rate (LBR). Clinical PR was calculated as the percentage of patients with gestational sac/s detected after ET. MR was calculated as the percentage of miscarriages up to the 20th week of gestation of the total of patients with clinical PR, and LBR was calculated as the number of deliveries of at least 1 live-born infant divided by the total number of patients undergoing ET.

### 2.6. Statistical Analysis

Data are shown as the mean ± standard deviation (SD) or percentages (as defined in the Section 2.5), and minimal and maximal values were found. For continuous variables, analysis of variance (ANOVA) was performed; for categorical ones, the chi-squared test was used. Statistical analyses were performed with Statistical Package for Social Sciences 23 (SPSS; Chicago, IL, USA) software. A *p* value < 0.05 was considered statistically significant.

## 3. Results

Analysis comprised 724 couples divided into two groups: The study group included those cycles in which MACS selection was performed after the gradient centrifugation procedure (*n* = 366); in the control group, only gradient centrifugation for sperm selection was performed (*n* = 358). There were no significant differences in the initial level of SDF or patient demographic characteristics between the MACS and control groups, as summarized in Table 1. The numbers of collected oocytes and Metaphase II oocytes were similar in both groups (*p* = 0.36 and *p* = 0.31; Table 1). The evaluation of reproductive outcomes for all patients enrolled in the study is shown in Table 2. Mean fertilization rate in the MACS technique group was 75.1%, and 73.3% in the control group, with no statistically significant differences between groups (*p* = 0.13; Table 2). By contrast, significant differences were found between the MACS and control groups in the clinical pregnancy rate per cycle (60.7% vs. 51.5%, respectively; *p* = 0.014), miscarriage rate (14.7% vs. 20.6%, respectively; *p* = 0.034), and live-birth rate (47.4% vs. 31.2%, respectively; *p* = 0.001). To know if those significant differences were maintained in the different ART procedures, reproductive outcomes were separately analyzed in both groups in the 3 different procedures: preimplantation genetic testing for aneuploidy (PGT-A) ICSI cycles, autologous ICSI cycles, and oocyte-donation ICSI cycles.

In the PGT-A cycles (*n* = 126 in the MACS group vs. *n* = 116 in the control group), no statistically significant differences were observed between the two groups in fertilization rate (76.0% vs. 76.8%, respectively; *p* = 0.201), pregnancy rate (60.4% vs. 50.6%, respectively; *p* = 0.121), miscarriage rate (15.1% vs. 11.4%, respectively; *p* = 0.307) or live-birth rate (43.4% vs. 31.6%, respectively; *p* = 0.127) (Table 3).

When autologous ICSI cycles were analyzed (*n* = 126 in the MACS group vs. *n* = 116 in the control group), results were equivalent between the MACS and control groups in fertilization rate (73.6% vs. 75.09%, respectively; *p* = 0.586) and pregnancy rate (52.2% vs. 50.0%, respectively; *p* = 0.424). However, strong significant differences were found when miscarriage rates (11.3% in MACS vs. 25.5% in control groups; *p* = 0.005) and live-birth rates (40.9% in MACS vs. 24.6% control groups; *p* = 0.03) were analyzed (Table 3).

Lastly, for those couples that had undergone oocyte-donation cycles, fertilization rate was similar in the MACS and control groups (76.85% vs. 76.95%, respectively; *p* = 0.750). Miscarriage rate was lower, but not statistically significant, in the MACS group compared to in the control group (17.9% vs. 22.5%, respectively; *p* = 0.247). Nonetheless, consistent significant differences were observed when pregnancy rate (69.6% in MACS vs. 53.9% in control groups; *p* = 0.013) and live-birth rate (51.8% in MACS vs. 29.4% control groups; *p* = 0.03) were analyzed (Table 3).

## 4. Discussion

This retrospective study assessed the effect of the MACS sperm-selection technique for patients with high levels of sperm DNA fragmentation undergoing ICSI on their clinical reproductive outcomes by assessing fertilization, pregnancy, miscarriage, and live-birth rates. Overall, results showed improvement in pregnancy, miscarriage, and live-birth rates when MACS sperm selection was performed in comparison to results from the control group. However, results were not the same when ART procedures were independently analyzed.

In PGT-A cycles, no significant differences were found in reproductive outcomes when comparing both groups. These procedures included strict selection (euploid embryo transfer) that may bypass the influence of the male factor in the results. Even so, a trend towards improvement in pregnancy and live-birth rates was observed in the MACS group, suggesting that the elimination of apoptotic sperm could have a beneficial effect in the development and subsequent implantation of euploid embryos. Associations between sperm-derived chromosomal abnormalities, and apoptosis and recurrent pregnancy loss were previously documented [28].

Autologous ICSI cycles showed strong significant improvement in reproductive outcomes when the MACS sperm-selection technique was performed, with a significant decrease in miscarriage rate and an increase in live-birth rate. These results suggested that high levels of sperm DNA fragmentation were closely related to an increase in miscarriages. These findings were consistent with those in many previous studies and meta-analyses [29,30,31,32,33], and refuted others that failed to confirm this correlation [34]. However, this variability in results could be partly due to the different origin and types of DNA strand breaks in sperm. In this sense, abortive apoptosis mainly originates from DNA double-strand breaks in sperm [11], and this type of DNA fragmentation is associated with an increase in miscarriage risk [35]. As the MACS procedure is based on the selective elimination of apoptotic cells, the sperm population isolated after this sperm-selection technique presumably had a significant reduction in sperm containing double-strand breaks and thereby reduced the chance that one of these spermatozoa would be selected for ICSI cycles.

When oocyte-donation cycles were analyzed, as in the previous procedures, no significant differences were found in fertilization rates between groups. Nonetheless, a significant increase in both pregnancy and live-birth rates was observed in the MACS group compared to those in the control group. These results are similar to those of previous studies [36,37] in which improvement in reproductive outcomes was also reported in oocyte-donation cycles for couples with an associated male factor. On the other hand, previous studies showed that the MACS procedure was unable to improve reproductive outcomes in oocyte-donation cycles. However, this study did not consider sperm DNA fragmentation in male inclusion criteria [23].

Although the MACS procedure may result in improved reproductive outcomes, published success is still scarce and uncertain. An explanation for these contradictory results may arise from the fact that detailed selection criteria for male subjects included in these studies are nonexistent or inefficient. Recently, published results demonstrated that sperm DNA fragmentation delayed embryo cleavage when donated oocytes were used for ICSI [38]. A novel retrospective study of 1602 pregnancies from IVF and ICSI cycles showed that, above a certain threshold, sperm with immature chromatin slightly increased the risk of early miscarriage [39]. In this sense, a previous article reported a decrease in miscarriage rate in ICSI cycles after MACS sorting of semen containing spermatozoa with high DNA fragmentation [40]. Additionally, there was a significant increase in the percentage of high-quality embryos and clinical pregnancies when MACS sperm selection was performed for 80 infertile couples with an underlying disturbing male factor who underwent ICSI when compared to the study’s control group [22]. Interestingly, when high levels of sperm DNA fragmentation were not used as a selection criterion, no advantage for MACS sperm selection was found [23].

Sperm DNA fragmentation can also be induced by epididymal oxidative damage caused by an increase in reactive oxygen species (ROS). There are different techniques to measure ROS production and/or concentration, although the great variability of the obtained results through the different techniques has made it impossible to reach a consensus on the reference values. In this study, we did not analyze oxidative damage, so we cannot rule out the influence of this mechanism on the increase in SDF. However, excessive ROS can induce DNA damage and activate apoptotic pathways in spermatozoa [41], which could be detected by the MACS technique.

The main limitation in this study was the fact that the data were retrospectively analyzed; a subsequent randomized controlled trial is needed to reduce any sources of bias. However, the large sample size used in our study (*n* = 724) made the results quite reliable and consistent.

In summary, the results presented in this retrospective study showed that patients with high levels of sperm fragmentation could benefit from the MACS technique to select the best sperm, and thereby improve embryo quality and live-birth rates.

## 5. Conclusions

The MACS sperm selection of human spermatozoa is a safe, easy, and suitable method for sperm preparations for ICSI use in a clinical setting. Results presented in this study indicated that the MACS selection technique may help to improve reproductive outcomes for autologous or oocyte-donation cycles in patients with high levels of sperm DNA fragmentation.

## Figures and Tables

**Figure 1 jcm-09-03976-f001:**
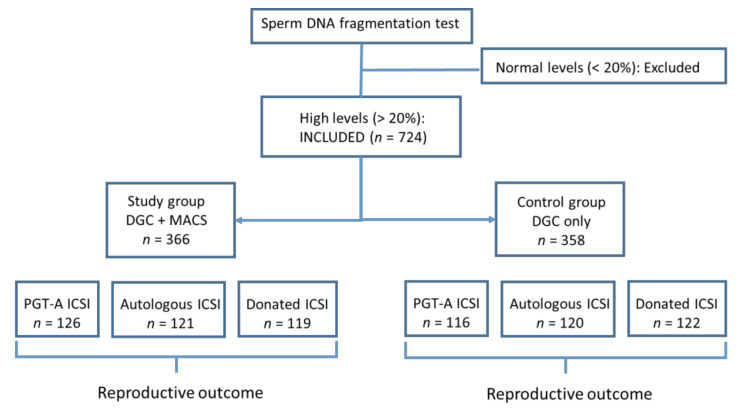
Inclusion and study groups.

**Table 1 jcm-09-03976-t001:** Main patient and cycle characteristic properties of cases included in the study.

	Study Group MACS (*n* = 366)	Control Group w/o MACS (*n* = 358)	*p*
Female age (year) ^a^	372 ± 3.6	36.7 ± 3.5	0.3
Male age (year) ^a^	40.0 ± 15.7	38.7 ± 15.4	0.28
SDF (%) ^a^	28.9 ± 8.2	29.6 ± 9.1	0.30
Total sperm count (mil) ^a^	89.1 ± 79.4	95.1 ± 87.2	0.15
Progressive motility (%) ^a^	35.2 ± 18.1	33.3 ± 16.7	0.12
Number of collected oocytes	11.3 ± 7.2	11.9 ± 6.8	0.36
Number of Metaphase II oocytes	9.7 ± 5.0	9.5 ± 5.1	0.31

Note: MACS, magnetic-activated cell sorting; SDF, sperm DNA fragmentation; ^a^ values expressed as mean ± standard deviation.

**Table 2 jcm-09-03976-t002:** Reproductive outcome in cycles studied.

	Study Group MACS (*n* = 366)	Control Group w/o MACS (*n* = 358)	*p*
Fertilization rate (%)	75.1	73.3	0.133
Pregnancy rate (%)	60.7	51.5	0.014
Miscarriage rate (%)	14.7	20.6	0.034
Livebirth rate (%)	47.4	31.2	0.001

**Table 3 jcm-09-03976-t003:** Reproductive outcome in studied cycles. Note: ICSI, intracytoplasmic sperm injection.

**Preimplantation Genetic Testing for Aneuploidy (PGT-A) Cycles**			
	**Study Group MACS (*n* = 126)**	**Control Group w/o MACS (*n* = 116)**	***p***
SDF (%)	28.8	30.1	0.168
Fertilization rate (%)	76	76.8	0.201
Pregnancy rate (%)	60.4	50.6	0.121
Miscarriage rate (%)	15.1	11.4	0.307
Live-birth rate (%)	43.4	31.6	0.127
**Autologous Oocyte ICSI Cycles**			
	**Study Group MACS (*n* = 121)**	**Control Group w/o MACS (*n* = 120)**	***p***
SDF (%)	30.7 ± 10.2	30.8 ± 10.8	0.958
Fertilization rate (%)	76.8	75.1	0.586
Pregnancy rate (%)	52.2	50.0	0.424
Miscarriage rate (%)	11.3	25.5	0.005
Live-birth rate (%)	40.9	24.6	0.03
**Oocyte-Donation Cycles**			
	**Study Group MACS (*n* = 121)**	**Control Group w/o MACS (*n* = 120)**	***p***
SDF (%)	27.7 ± 6.8	28.0 ± 7.2	0.959
Fertilization rate (%)	76.85	76.9	0.750
Pregnancy rate (%)	69.6	53.9	0.013
Miscarriage rate (%)	17.9	22.5	0.247
Live-birth rate (%)	51.8	29.4	0.03

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
