# Peer review of "Magnetic-Activated Cell Sorting (MACS): A Useful Sperm-Selection Technique in Cases of High Levels of Sperm DNA Fragmentation"

_jcm, 2020, doi:10.3390/jcm9123976_

Round 1
Reviewer 1 Report
I found this paper interesting but the following points need to be addressed:
- What are the other causes of SDF? Were these not evaluated in patients - if not why not? There is significant evidence in the literature suggesting that ART outcomes may be affected by other factors- expand on these.
- What was the basis of using a threshold of 20% for TUNEL?
- How does this group manage patients with raised SDF- are there any further investigations performed in this group eg screening for other andrological causes?
- There is no information on other confounding variables eg number of previous ART cycles, AMH etc- would these have influenced LBR-yes?
- Provide an algorithm on the management of SDF in this clinical setting- do you simply used MACS as your primary treatment modality?
- My concern is that although exclusion criteria are mentioned, these are poorly documented. We need to see more demographic data in the male- I am assuming that these patients would have undergone Urological evaluation?
- What are the specific indications to measure SDF in your unit and where is the evidence for this?
Author Response
Manuscript ID: jcm-1005297
Type of manuscript: Article
Title: Magnetic Activated Cell Sorting (MACS) Is a Useful Sperm Selection Technique in Cases of High Levels of Sperm DNA Fragmentation
Authors: Alberto Pacheco *, Arancha Blanco, Fernando Bronet, Maria Cruz, Jaime Garcia-Fernandez, Juan A. Garcia-Velasco
Reviewer 1
I found this paper interesting but the following points need to be addressed:
- What are the other causes of SDF? Were these not evaluated in patients - if not why not? There is significant evidence in the literature suggesting that ART outcomes may be affected by other factors- expand on these.
As we already mentioned in the manuscript (2nd paragraph Introduction), apart from testicular apoptosis, the other main causes of SDF in spermatocytes are defects in DNA remodeling during spermatogenesis, and epididymal oxidative damage. The analysis of remodeling defects is not currently standardized, and because of that, it was not performed in these patients. In the case of oxidative damage, there are different techniques to measure reactive oxygen species (ROS) although the great variability of the obtained results through the different techniques, has made it impossible to reach a consensus on the reference values of clinical use. However, excessive ROS can induce DNA damage and also activate apoptotic pathways in spermatozoa [Agarwal A. et al 2020, Duta S et al 2019], that could be detected by MACS technique. We have added this comment in Discussion section, and included a reference,
- What was the basis of using a threshold of 20% for TUNEL?
This threshold value was determined according to our own studies using semen donor with proven fertility, although other authors have published the same or very similar threshold values, as we mentioned in the manuscript (Introduction section).
We have described it more clearly in the revised manuscript.
- How does this group manage patients with raised SDF- are there any further investigations performed in this group eg screening for other andrological causes?
Patients with altered DNA fragmentation or severe male defects (severe oligo and/or asthenozoospermia) are referred to our andrologist for evaluation in our institution prior to assisted reproductive treatment, for further investigations if needed and rule out other urologic pathologies, such as varicocele, orchitis, epididymitis….
We have included this protocol in the revised manuscript.
- There is no information on other confounding variables eg number of previous ART cycles, AMH etc- would these have influenced LBR-yes?
We agree with the reviewer that this information related to the background of the couple or the female factor is not proviede. However, considering that we are discussing male factor approach, and that given the large sample size distribution would be even, we did not consider to include it to avoid extending the length of the manuscript with unrelevant information. However, if needed, we can provide it.
- Provide an algorithm on the management of SDF in this clinical setting- do you simply used MACS as your primary treatment modality?
In cases of high level of SDF, and if it is maintained in semen samples after urological investigation and/or treatment, this (MACS) is our primary treatment of sperm selection technique for ICSI cycles.
- My concern is that although exclusion criteria are mentioned, these are poorly documented. We need to see more demographic data in the male- I am assuming that these patients would have undergone Urological evaluation?
Thank you for your comment. We have included more detail of exclusion criteria in the revised manuscript
- What are the specific indications to measure SDF in your unit and where is the evidence for this?
The indications for SDF measurement in our unit are seminal defects (especially asthenozoospermia), lifestyle and environmental factors associated with SDF (obesity, smoking, exposure to chemicals substances…), male age > 45 yo and previous ART failures in other centers, without a female factor associated. All of these indications have been previously documented:
Petersen CG, Mauri AL, Vagnini LD, Renzi A, Petersen B, Mattila M, et al. The effects of male age on sperm DNA damage: an evaluation of 2,178 semen samples. JBRA Assist Reprod 2018;22:323-30.
Gallegos G, Ramos B, Santiso R, Goyanes V, Gosálvez J, Fernández JL. Sperm DNA fragmentation in infertile men with genitourinary infection by Chlamydia trachomatis and Mycoplasma. Fertil Steril 2008;90:328-34
Yang Q, Zhao F, Hu L, Bai R, Zhang N, Yao G, et al. Effect of paternal overweight or obesity on IVF treatment outcomes and the possible mechanisms involved. Sci Rep 2016;6:29787.
Calogero A, Polosa R, Perdichizzi A, Guarino F, La Vignera S, Scarfia A, et al. Cigarette smoke extract immobilizes human spermatozoa and induces sperm apoptosis. Reprod Biomed Online 2009;19:564-71.
Jerre E, Bungum M, Evenson D, Giwercman A (2019). Sperm chromatin structure assay high DNA stainability sperm as a marker of early miscarriage after intracytoplasmic sperm injection. Fertil Steril. 2019; 112(1):46-53
Reviewer 2
Magnetic Activated Cell Sorting (MACS) Is a Useful Sperm Selection Technique in Cases of High Levels of Sperm DNA Fragmentation by Alberto et al. analysed an important area of ART- sperm selection by MACS and its outcome. The tables (sub-categories) may be combined into a single table and avoid giving a separate column for P-value (it can be denoted with an asteric as significant or non-significant at the bottom of the table) Since the figure is a repetition of the tables it can be avoided
Answer: Thank you for your comments. According to your suggestion, we have unified the different treatment subcategories in a single table. However, we have not eliminated the column from the p-value, since we consider that the number gives more information than a simple symbol.
Following your comments, the Figure 2 has been avoided.
Reviewer 2 Report
Magnetic Activated Cell Sorting (MACS) Is a Useful Sperm Selection Technique in Cases of High Levels of Sperm DNA Fragmentation by Alberto et al. analysed an important area of ART- sperm selection by MACS and its outcome. The tables (sub-categories) may be combined into a single table and avoid giving a separate column for P-value (it can be denoted with an asteric as significant or non-significant at the bottom of the table)
Since the figure is a repetition of the tables it can be avoided
Author Response

(The authors gave the same response as above.)
